# Pile-Ups Formation in AFM-Based Nanolithography: Morpho-Mechanical Characterization and Removal Strategies

**DOI:** 10.3390/mi13111982

**Published:** 2022-11-15

**Authors:** Paolo Pellegrino, Isabella Farella, Mariafrancesca Cascione, Valeria De Matteis, Alessandro Paolo Bramanti, Lorenzo Vincenti, Antonio Della Torre, Fabio Quaranta, Rosaria Rinaldi

**Affiliations:** 1Department of Mathematics and Physics “Ennio De Giorgi”, University of Salento, Via Monteroni, 73100 Lecce, Italy; 2Institute for Microelectronics and Microsystems (IMM), CNR, Via Monteroni, 73100 Lecce, Italy; 3STMicroelectronics S.r.l., System Research and Applications (SRA) Silicon Biotech, Lecce Labs, Via Monteroni, 73100 Lecce, Italy

**Keywords:** pile-ups characterization, nanomechanics, atomic force-nanolithography, Pulse-Atomic Force Nanolithography, atomic force microscopy

## Abstract

In recent decades, great efforts have been made to develop innovative, effective, and accurate nanofabrication techniques stimulated by the growing demand for nanostructures. Nowadays, mechanical tip-based emerged as the most promising nanolithography technique, allowing the pattern of nanostructures with a sub-nanometer resolution, high reproducibility, and accuracy. Unfortunately, these nanostructures result in contoured pile-ups that could limit their use and future integration into high-tech devices. The removal of pile-ups is still an open challenge. In this perspective, two different AFM-based approaches, i.e., Force Modulation Mode imaging and force-distance curve analysis, were used to characterize the structure of pile-ups at the edges of nanogrooves patterned on PMMA substrate by means of Pulse-Atomic Force Lithography. Our experimental results showed that the material in pile-ups was less stiff than the pristine polymer. Based on this evidence, we have developed an effective strategy to easily remove pile-ups, preserving the shape and the morphology of nanostructures.

## 1. Introduction

During the last decades, the rapid advance in nanomanufacturing techniques has enabled the production of high-quality nanostructures; including with complex shapes, on different substrates, and even on a large scale, for integration into high-tech devices. Yet, the demand for new nanofabrication techniques, ever more efficient and accurate, is unceasing [1,2,3,4]. In this framework, the Tip-Based Nanofabrication (TBN) approaches [2,5] have emerged as the most promising nanolithography methods due to their versatility, flexibility, low cost, accuracy, and nanoscale resolution [5]. Up to now, several types of TBN methods have been developed, such as Local Anodic Oxidation (LAO) [6] or thermal [7], electric [8], dip pen [9], and mechanical lithography [10]. Particularly, mechanical lithography (m-TBN) plays an important role in the nanofabrication panorama since it permits the manipulation of materials with a sub-nanometer resolution by applying a force on the sample surface with many different operation modes [11,12,13,14]. The m-TBN techniques have been used to pattern 2D, 2.5D, and 3D nanostructures using conventional AFM instruments. Still, some detrimental effects, such as edge irregularities and high pile-up materials, affected the quality of the produced nanostructures. Improvements in the nanostructure’s quality were achieved by coupling conventional m-TBN techniques with additional energy sources such as high tip temperature [15], tip rotation [16], and bias voltage [8]. Unfortunately, these hybrid processes require very elaborate equipment and, often, multi-step approaches, resulting in an increase in complexity and working time. To overcome these limits, we have developed an innovative m-TBN technique based on Atomic Force Microscopy (AFM) nanoindentation and termed Pulse-Atomic Force Lithography (P-AFL) [17], with which we can easily realize nanogrooves on a thin polymer layer, with high accuracy and reproducibility. P-AFL enables the fabrication of nanostructures with high resolution in the *xy*-plane and a continuously variable depth profile (*z*-axis), or a constant one, upon the appropriate setting of the pulse amplitude (Setpoint), the pulse width, and the distance between subsequent indentations (Step). 

Therefore, P-AFL enables surface patterning with high-accuracy nanometric spatial resolution, and easy depth-tuning, just in a single pass, overcoming one of the main drawbacks of conventional m-TBM approaches represented by their laboriousness. Unfortunately, the formation of high pile-ups is still observed at the nanogroove edges due to the mechanical displacement of material from the groove during nanolithography (Figure 1a). Indeed, we observed asymmetric pile-ups building up aside the nanogrooves, higher at the bottom side (i.e., far from the end of the cantilever) than at the upper. The height of these bulges increases with the depth of channels, and the hillocks on both sides were always higher than the depth of the nanogrooves at the same point. We speculated that these bulges might not be homogeneous and compact due to fractures or cracks inside (Figure 1b). This assumption was partially corroborated by the results of preliminary stiffness estimation, which showed that Young’s modulus (*E*) value of PMMA decreased in correspondence with pile-ups concerning the non-patterned polymer [17]. Although the formation of pile-ups at the edges of nanostructures patterned with most of m-TBNs has already been observed, to date, few experimental works have been devoted to the morphological and mechanical characterization of these bulges. As far as we know, a very preliminary characterization of the pile-up stiffness has been performed by Yan and co-workers [18] and our group [17]. To definitively validate our previous hypothesis, we carefully characterized the PMMA pile-up stiffness through two different AFM experiments: Force Modulation Mode (FMM) imaging and force-distance (FD) curve acquisitions, which enable respective mapping of the surface mechanical characteristics at a nanoscale resolution [19,20,21,22,23,24] and locally measuring the elastic properties of the material surface [25,26,27]. To our knowledge, this paper is the first that concerns a complete characterization of pile-up morphomechanical properties and an easy and very effective protocol for completely removing them without damaging the non-patterned PMMA. 

## 2. Materials and Methods

### 2.1. Substrate Preparation

The substrate used was a standard 4-inch (100) silicon wafer coated with a 0.5 μm thermally grown silicon dioxide layer. First, the wafer was cleaved into 1 cm × 1 cm pieces, then cleaned by sequential sonication baths in acetone and 2-propanol (both purchased from Sigma-Aldrich, St. Louis, MO, USA). The samples were rinsed under deionized water flow and dried out under nitrogen gas flow. A thin layer of PMMA A4 (950 kDa, in solvent anisole, MicroChem, Berlin, Germany) was spun for 30 s at 4000 rpm using a semiautomatic spinner DELTA 80T (SUSS MicroTec Corp, Garching, Germany); finally, the samples were baked on a hotplate for 90 s at 180 °C to remove anisole solvent. The thickness of the PMMA layer was measured with an Alpha-Step P6 profilometer (KLA-Tencor Corporation, Milpitas, CA, USA).

### 2.2. Instrumentation for Nanolithography and Morphological/Mechanical Characterization

The nanolithography process and morphological characterization of the patterned surface of samples have been carried out at ambient conditions using the AFM NTEGRA (NT-MDT Spectrum Instruments, Moscow, Russia). P-AFL experiments have been performed using DCP20 probes (NT-MDT Spectrum Instruments, Moscow, Russia), which have a V-shaped cantilever with a resonant frequency equal to 510 kHz and a nominal spring constant (*k*) ranging from 40 to 110 N/m. The latter has been accurately quantified using the thermal noise method before each P-AFL test [28]. The DCP20 tips have also been employed to acquire force-distance curves on pile-ups and pristine PMMA, to estimate their stiffness. 

Conversely, the CSG01 probe (NT-MDT Spectrum Instruments, Moscow, Russia) has been adopted for Force Modulation Mode characterizations on untreated and patterned PMMA samples. The CSG01 cantilevers have a resonant frequency of ~9.8 kHz and a nominal elastic constant of 0.03 N/m; the tip at the apex of the cantilever has a tetrahedral shape, with a typical tip curvature radius smaller than 10 nm, and a tip cone angle of less than 20°.

The high-resolution topographic images have been acquired in semi-contact error mode utilizing NSG01 tips (NT-MDT Spectrum Instruments, Moscow, Russia). These probes are characterized by a rectangular-shaped cantilever with a tip curvature radius of about 6 nm and a nominal spring constant of 5 N/m. After the acquisition, the topographic images were analyzed to evaluate the roughness. Specifically, it was expressed in terms of root-mean-square surface roughness (*R_q_*), which was quantified by the following equation:(1)$$R_{q}=\sqrt{\frac{1}{n}\sum_{i=1}^{n}z_{i}^{2}}$$
where *n* is the number of data points, and *z_i_* is the height deviation of the *i*-th point from a mean line, the latter making the arithmetic sum of all *z_i_* equal to zero [29]. This analysis was performed over ten 5 µm × 5 µm areas, selected based on the high-resolution topographic images on pristine PMMA and along the inner central part of nanogrooves; therefore, *R_q_* was expressed as mean value ± SD.

Prior to *R_q_* measurements, each topographical acquisition was digitally treated with a second-order plane fit and with a second-order flattening to suppress bow and tridimensionality effects. Furthermore, the topography of nanochannels was also analyzed to quantify the geometrical parameters of the pile-ups. Specifically, depth, width, and height have been calculated as mean value ± SD by practicing 20 cross-sections *per* acquisition.

### 2.3. Nanolithography Protocol

A set of 5 μm-long nanochannels have been patterned on PMMA by P-AFL, according to the optimized procedure presented in our previous paper [17], setting the following nanolithography parameters: setpoint of 5 nA (force equal to (9.4 ± 0.2) µN), Step (distance between the following indentation) equal to 10 nm and pulse width 10 ms.

### 2.4. Force Modulation Mode Investigation

FMM characterization has been performed in ambient conditions at an actuation frequency of 6.7048 kHz and a lock-in amplifier gain coefficient set at its maximum value, acquiring the height, amplitude, and phase images with a setpoint of 2.0 nA, a gain of 0.9, a rate of 0.8 Hz, and a 256 × 256 points resolution.

### 2.5. PMMA Stiffness Estimation

The force-spectroscopy investigation has been performed by acquiring FD curves on pristine PMMA and pile-ups risen at the nanochannel edges during the P-AFL process to quantify the stiffness expressed in terms of Young’s modulus (*E*). The latter has been obtained by fitting the indentation curve with the Sneddon model [27,30]:(2)$$F=\frac{h^{2}2\;E\;tan\alpha }{\pi \left(1-\nu^{2}\right)}$$
in which *F* indicates the force, *h* is the displacement of the indenter, *ν* is the Poisson’s ratio of the sample, *α* is the half-angle of the conical tip, and *E* is evaluated as the best fit parameter. The FD curves have been obtained on 500 nm × 500 nm areas, setting a grid of 15 × 15 points within each area, thereby recording 225 FD curves per topographic acquisition. The *E* values were obtained from the Gaussian fitting of Young’s modulus distribution and reported as mean ± SD.

### 2.6. Pile-Ups Removal Protocol

According to methods reported in Section 2.1, other substrates were specially prepared to optimize the removal protocol of pile-ups formed after carving 3 µm long nanogrooves patterned as described in Section 2.3. Those samples were immersed in a 1:4 (*v*/*v*) solution of Methyl Isobutyl Ketone (MIBK) (MicroChem, Berlin, Germany) and 2-propanol (IPA) for different exposure times (0 s, 10 s, 20 s, 30 s, 60 s, 90 s). Then, the samples were rinsed with deionized water and dried out under a nitrogen stream.

### 2.7. Software

The NOVA_PX software has been used to perform the nanolithography test, the FMM characterization, the force-spectroscopy analysis, and all the morphological characterizations. The AFM images, the force-spectroscopy analysis, the indentation curves acquisition, and the FMM images were analyzed using the Image Analysis P9 software. IA-P9 and NOVA_PX software are from NT-MDT Spectrum Instruments, Moscow, Russia. The data were analyzed and plotted by Origin Pro v8 (Origin-Lab Corporation, Northampton, MA, USA).

### 2.8. Statistical Analysis 

The results presented in this work are given as mean values and associated standard deviation (±SD). The differences among data were analyzed through ANOVA multiple comparisons. The differences were statistically significant when *p* < 0.01.

## 3. Results

Prior to patterning the nanochannels by P-AFL, the thickness of the PMMA substrate was measured by a profilometer, resulting equal to ~70 nm, and the surface of the substrate was imaged using AFM equipped with a CSG01 probe in contact mode. The topographic AFM acquisitions showed that the PMMA surface was sufficiently smooth and homogeneous (Figure 2); this observation was confirmed by the superficial structural analysis that led to quantifying an *R_q_* value equal to (0.32 ± 0.08) nm.

These substrates were patterned using the P-AFL lithography [17] protocol, and the obtained nanostructures were visualized at high resolution using AFM in semi-contact error mode (Figure 3a–c). The morphometric analysis of nanochannels (Figure 3c,d) reveals a highly regular and homogeneous V-shaped geometry. In particular, the nanogrooves exhibited a constant depth and width equal to (48.88 ± 1) nm and (114.2 ± 4.8) nm, respectively. In addition, the *R_q_* estimated on the inner part of the nanochannels was (0.73 ± 0.1) nm. As already observed [17], the nanogrooves are asymmetrically contoured by PMMA bulges, whose heights were (47.11 ± 0.7) nm (Side-1) and (37.16 ± 0.5) nm (Side-2) (Figure 3).

The observed difference in bulge height could be explained by assuming that, during indentation, the DCP20 probe bents over, inducing the asymmetric accumulation of PMMA at the groove edges. Furthermore, the sum of their heights was surprisingly greater than the nanochannel depth. This experimental finding led us to suppose that PMMA was not compact and homogeneous in the pile-ups, rather presenting internal fractures or cracks. To check this hypothesis, FMM acquisitions and FD analysis were carried out.

FMM is a powerful technique in which the AFM tip scans the sample surface in contact mode while an additional modulated voltage is fed to the z-section of the piezo-scanner, producing vertical, sinusoidal oscillations at a fixed frequency. The consequent, additional cantilever deflection depends on the viscoelastic properties of the sample. Using a lock-in amplifier, the amplitude and phase responses of the cantilever vibration are detected and recorded as a function of the tip position. Nonetheless, the surface topography signal is simultaneously acquired, and even slight differences in the sample elasticity can be appreciated by an FMM image [8]. 

The height channel of FMM acquisitions (Figure 4a) showed a typical nanogroove contoured by two asymmetric pile-ups. All the measured heights confirmed the values from the previous contact mode characterization (Figure 3d). 

The amplitude channel (Figure 4b) showed significantly lower amplitude in pile-ups than in pristine PMMA—more evidently in Side-1 (1.7 nA) than in Side-2 pile-ups (3.9 nA), both versus 7.5 nA of PMMA (Figure 4e). In addition, the phase signal increased over the pile-ups (Figure 4c,f). The contemporary amplitude decreasing and phase increasing in the bulges, with respect to pristine PMMA, indicated that the accumulated material in pile-ups is less stiff than in the non-patterned substrate.

To quantify the change of PMMA stiffness in pile-ups, a careful Force-Spectroscopy investigation was carried out. The distribution of the E values across the sample surface area was plotted and converted into a 2D elasticity map (Figure 5a,b). The map shows qualitatively, but clearly, the change in stiffness passing from the PMMA region (yellow) to the pile-up one (blue) (Figure 5b).

Within these two main regions, the stiffness appeared almost homogeneous. The Sneddon analysis of FD curves on pristine PMMA (Figure 6a) resulted in the estimated E modulus equal to (5.22 ± 0.01) GPa, about 20% higher with respect to the E of pile-ups on Side-1 and Side-2, equal to (4.06 ± 0.05) GPa and (4.19 ± 0.04) GPa, respectively (Figure 6b,c). Finally, Young’s modulus of the pile-ups on Side-1 and Side-2 were comparable and statistically significantly different (*p* < 0.01).

The correctness of our hypothesis on the less compact and homogeneous internal structure of polymeric bulges, corroborated by the results obtained with Force Modulation imaging and FD analysis, led us to think that a treatment similar to the development of resist after exposure to an electron beam could be borrowed and applied to remove pile-ups effectively. In this perspective, we immersed the nanopatterned samples in MIBK/IPA solution (1:4 *v*/*v*), commonly used as a developer of PMMA after EBL exposure, for 10, 20, 30, 60, and 90 s. The effect of this treatment on nanostructures was analyzed using AFM in semi-contact mode (Figure 7). It is evident from the topographic acquisitions that the MIBK/IPA treatment induced a reduction in the height of the pile-ups (Figure 7). After an exposure time of 30 s, the pile-ups appeared almost completely removed. Increasing the exposure time to 90 s showed an increase in the width and depth of the channel, as well as a change in its geometry, passing from a V- to a rectangular-shaped profile.

These qualitative speculations were validated through quantitative analysis of morpho-structural parameters, describing the nanogrooves at different time points of MIBK/IPA exposure (Figure 8). Specifically, the topographical analysis revealed that 30 s of treatment was sufficient to almost completely remove the pile-ups; in fact, the height of the Side-1 pile-up decreased by ~94%, equaling ~97% when the exposure time was extended to 90 s (Figure 8a). A similar trend was observed for the Side-2 pile-up; nevertheless, after 30 s of MIBK/IPA treatment, the rate of decrease was less than 90% (Figure 8a). This might be justified by assuming that the lower pile-up material is more compact and, therefore, requires a longer time to be dissolved. The MIBK/IPA treatment also affects the shape and depth of the channels, as shown by the characterization of geometrical channel parameters at different time points (Figure 8b,c). The channel depth increased almost linearly with an exposure time of up to 30 s. Then it settled around ~70 nm, the thickness of the PMMA layer (Figure 8b). Regarding the channel width, a rapid increase was observed in the first 30 s, followed by a slower one in the successive 60 s. In this regime, while the depth remained almost unchanged because the silicon substrate was reached, the width slowly increased due to the channel wall degradation by the MIBK/IPA solution. The P-AFL patterning procedure induces a plastic deformation of PMMA, leaving the inner walls and the bottom of the channel as cracked as the pile-ups. The more exposed layers of the whole channels are more dissolvable in the solution regarding the non-deformed PMMA and take a few seconds (20 ÷ 30 s) to be removed. This causes the geometric profile of the channel to change, assuming a rectangular shape, as visible in Figure 7i,j.

The study on the effect of MIBK/IPA treatment was completed by analyzing the roughness parameter (*R_q_*), quantified in the middle part of nanochannels at different exposure times (Figure 8d). The *R_q_* dramatically increased after 10 s but then decreased back to a value comparable to the initial value after 90 s of treatment. We guessed that the material of the groove’s internal sidewalls, which only partially dissolved during the first 10 s, likely re-deposited at the bottom of the nanogroove, worsening its roughness. Keeping the sample immersed for a longer time is more effective in recovering the roughness of the untreated pattern.

Regarding the effect of MIBK:IPA treatment on the non-patterned PMMA surface, the roughness of the non-patterned polymer surface after every treatment time point was measured by analyzing the AFM images. The roughness analysis revealed a slight increase in the PMMA *R_q_* values, passing from (0.32 ± 0.03) nm before the cleaning, up to (0.39 ± 0.02) nm after 90 s of immersion in the MIBK:IPA solution. The height of the PMMA layer, obtained by measuring the distance from the PMMA surface to the channel bottom after 90 s of treatment, remains equal to ~70 nm, the pristine value of the PMMA layer, indicating no delamination or reduction in the height of the polymer effect, induced by the pile-ups treatment solution. 

Considering the reported results, it is possible to conclude that the prosed protocol effectively removes the pile-up bulges generated by AFM-based lithography without damaging the non-patterned surface. Under closer analysis, the treatment shows the drawback of slightly changing the channel geometry. The change in the channel morphology could be usefully exploited for those applications where well-shaped nanostructure walls are required, such as nanofluidics.

## 4. Conclusions

In light of these experimental findings, our hypothesis about the internal structure of pile-ups appears reasonable. The bulges only apparently contain more material than that removed from the channel during the mechanical-based lithography. The reduction in Young’s Modulus suggests that their internal structure (and, therefore, the material density) is different regarding non-patterned PMMA. Combined with an accurate FMM analysis, the decrease in the bulges’ stiffness provided the starting point to develop a simple and effective protocol to remove the pile-ups without damaging the lithographed pattern. The prolonged exposure to the proposed treatment induces a slight change in the depth and width of the carved channels. Nevertheless, these changes, together with a slight enlargement in the channel base, can be exploited for the future use of patterned nanogrooves in several applications, such as nanofluidics. 

## Figures and Tables

**Figure 1 micromachines-13-01982-f001:**
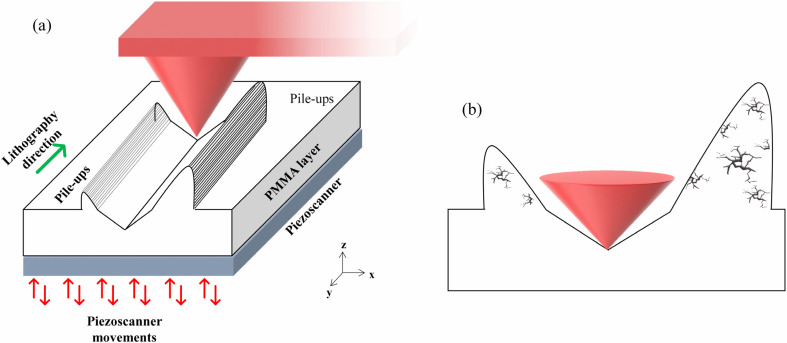
(**a**) Schematic representation of the pile-up formation during the P-AFL process on a thin PMMA film; the green arrow indicates the nanolithography direction, while the red arrows indicate the movement, in the z-direction, of the AFM piezo-scanner; (**b**) Sketch of the PMMA pile-up with cracks inside. In both images, the red cone shows the AFM tip.

**Figure 2 micromachines-13-01982-f002:**
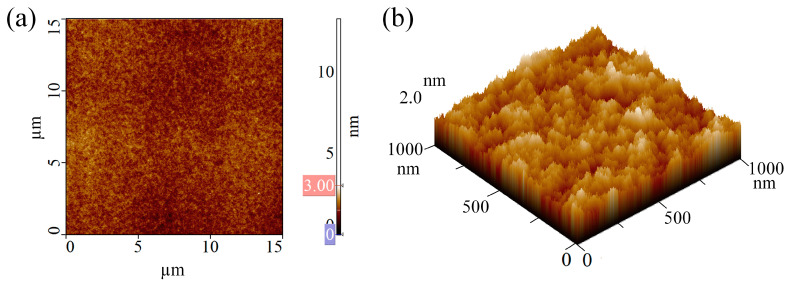
AFM images of pristine PMMA: (**a**) 2D and (**b**) 3D AFM topographical acquisitions in the SensHeight channel.

**Figure 3 micromachines-13-01982-f003:**
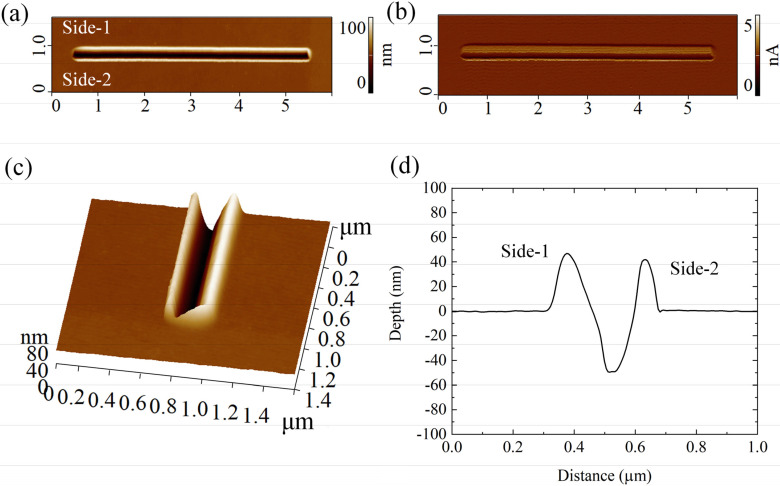
AFM topographical analysis performed on nanogrooves patterned on PMMA with P-AFL technique: 2D AFM topographical acquisitions in the height (**a**) and deflection (**b**) channels; 3D AFM reconstruction (**c**), and representative cross-section (**d**) of the nanochannel.

**Figure 4 micromachines-13-01982-f004:**
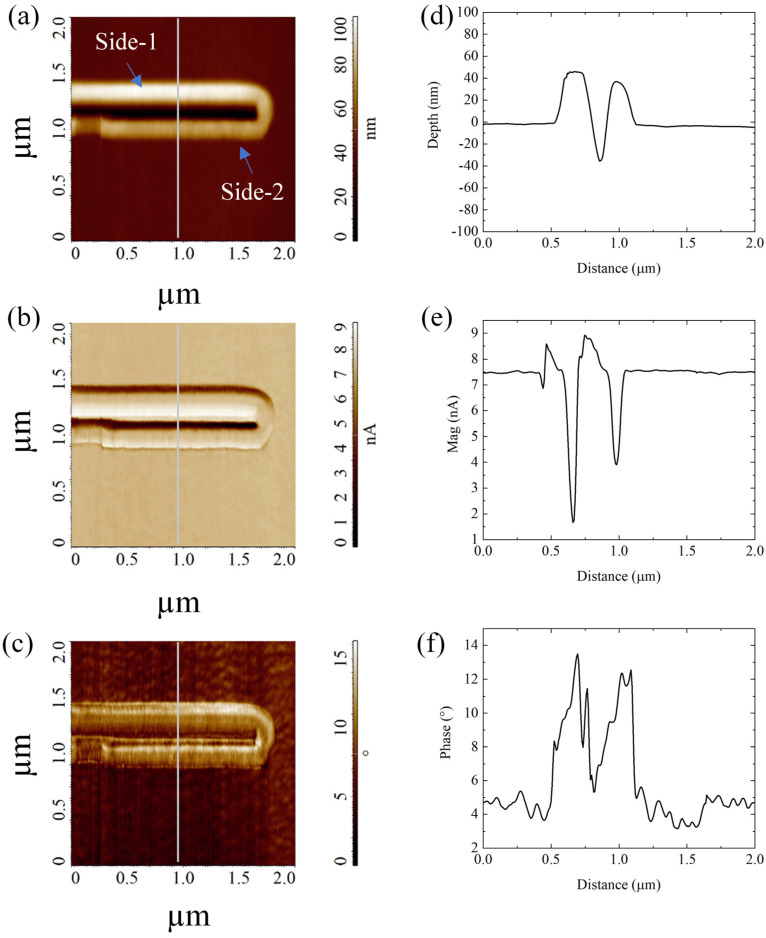
FMM images of a nanogroove and its pile-ups patterned on PMMA by the P-AFL method, with corresponding cross-section profiles acquired in the middle of the AFM image (grey bars). The image obtained in height (**a**), amplitude (**b**), and phase (**c**) channels showed a typical nanogroove contoured by two asymmetric pile-ups. In (**d**–**f**), the representative cross-section profile acquired on Height, Amplitude, and Phase images, respectively, were reported.

**Figure 5 micromachines-13-01982-f005:**
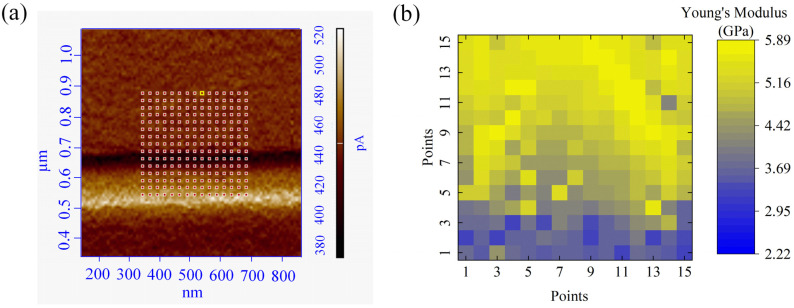
(**a**) Representative AFM topographic images, in the Magnitude channel, of a nanogroove area on which the FD curves were acquired. The red dots are the FD acquisition sites. (**b**) Qualitative elasticity map shows the elastic modulus values distribution on the selected sample area: each pixel in the map corresponds to a red dot in (**a**).

**Figure 6 micromachines-13-01982-f006:**
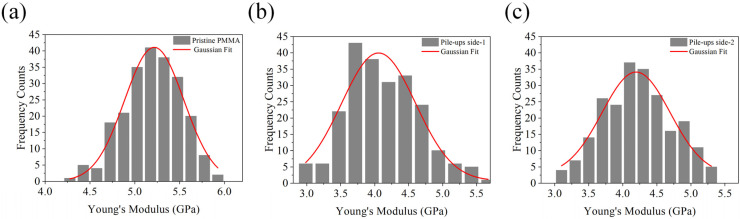
Young’s modulus distributions with Gaussian fit functions (red line) evaluated on 225 force-distance curves acquired on pristine PMMA (**a**) and pile-ups Side-1 (**b**) and Side-2 (**c**). Performing the ANOVA test, the mean values were statistically significant (*p* < 0.01).

**Figure 7 micromachines-13-01982-f007:**
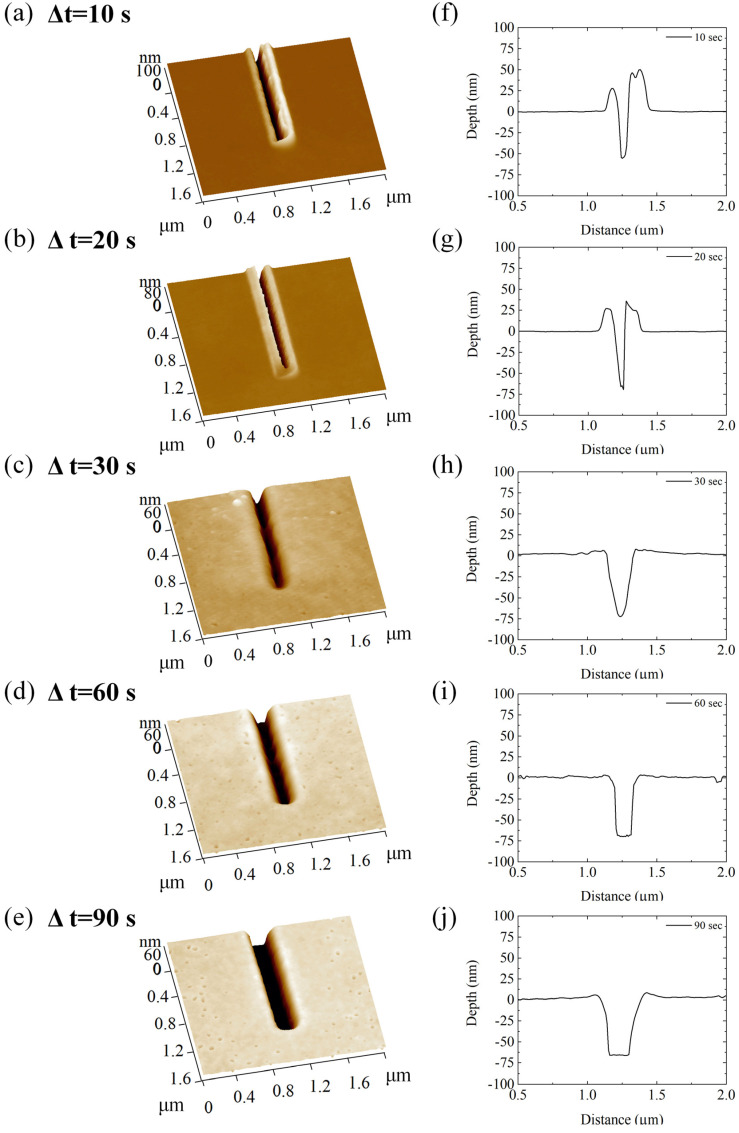
AFM topography imaging performed in semi-contact mode on nanogrooves sculpted on PMMA substrates. The acquisitions were carried out after treatment with MIBK and 2-propanol solution (1:4, *v*/*v*) at different exposure times: 10 s (**a**), 20 s (**b**), 30 s (**c**), 60 s (**d**), and 90 s (**e**). In (**f**–**j**), representative cross-sections of channels are represented.

**Figure 8 micromachines-13-01982-f008:**
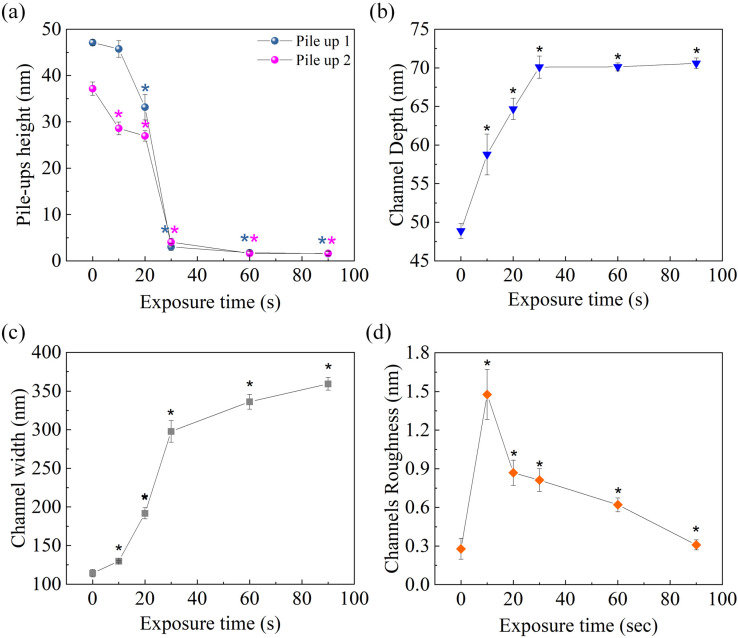
Variation of morpho-structural nanochannels parameters after treatment with MIBK and 2-propanol solution (1:4, *v*/*v*) at different exposure times: 10 s, 20 s, 30 s, 60 s, and 90 s. Change in pile-up height (**a**), channel depth (blue triangles) (**b**), width (gray squares) (**c**), and roughness (orange diamonds) (**d**). In the graphs, * indicates the statistical significance, evaluated using the *t*-Student test concerning the control value corresponding to t = 0 s (which refers to the sample untreated and reported in Figure 3).

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
