# Peer review of "Pile-Ups Formation in AFM-Based Nanolithography: Morpho-Mechanical Characterization and Removal Strategies"

_micromachines, 2022, doi:10.3390/mi13111982_

Round 1
Reviewer 1 Report
The manuscript is clearly written, methods and results are adequately described. However, in my opinion the amount of work (experiments, findings) is not extensive enough for a full article. It is more suitable as a short communication.
Additional comments are:
1. Line 46-48: Authors claim that the existing methods are laborious and complicated. In the present method of characterization and removal, the authors use similar AFM techniques. Therefore, a clarification is needed to support why the present methods are significantly simpler, inexpensive, less labour-intensive than previous methods.
2. Grammatical errors: Lines 16, 23
Author Response
The authors provided the point-by-point responses in the attached file.

Reviewer 2 Report
The authors showed experimental demonstration of pile up removal during mechanical AFM-based lithography process on a PMMA coated surface SiO2/Si substrate. The manuscript is well written and described the process elaborately.
1. The authors need to polish the Figures in the manuscript. The AFM topography figures (Fig. 2, 3, 7) need tick marks and scales (dimension). So as Fig. 7 for depth-distance curves (x-scale dimension).
2. Why the channel roughness increases first from 0.3 to 1.8 (Fig. 8 (d)) and then deceases? Please discuss in brief.
3. Is there any visible physical change in the sample surface after cleaning? Or any chemical change was observed? Please elaborate in the manuscript.
4. It appears that when the channel depth saturates (Fig. 8 (b)), the channel width starts to expand (Fig. 8(c)). Is there any control to stop the expansion according to the requirement, to stop unnecessary etching?
5. The authors published an article in Nanomaterials (Basel). 2022 Mar; 12(6): 991 where they deposited SiN/Sio2 and discussed the issues of pile up and its removal.
I would request the authors to present a qualitative and quantitative analysis of two removal processes and compare the results. Please elaborate why one is superior than another or user friendly.
5. What is the dependence of scanning rate vs. pile removal rate?
After answering the queries above, I would recommend the revised manuscript for publication.
Author Response
The authors provided point-by-point responses to the Reviewer comments in the file in attachment.

Round 2
Reviewer 1 Report
The authors have addressed the grammatical and technical question. However, the adequacy of data has not yet been sorted. The authors may communicate with the editor regarding this.
Reviewer 2 Report
Dear Authors,
I would like to thank you for answering the queries satisfactorily and for the required modifications in the manuscript.
best wishes